# Complementarity and 'Resource Droughts' of Solar and Wind Energy in Poland: An ERA5-Based Analysis

**Jakub Jurasz** [1,2,*] , **Jerzy Mikulik** [1] , **Paweł B. Dąbek** [3] , **Mohammed Guezgouz** [4] and **Bartosz Kaźmierczak** [2]

1    Faculty of Management, AGH University, 30-059 Cracow, Poland; jmikulik@zarz.agh.edu.pl
2    Faculty of Environmental Engineering, Wroclaw University of Science and Technology,
     50-377 Wroclaw, Poland; bartosz.kazmierczak@pwr.edu.pl
3    Institute of Environmental Protection and De-velopment, Wrocław University of Environmental and Life
     Sciences, Grunwaldzki Sq. 24, 50-363 Wrocław, Poland; pawel.dabek@upwr.edu.pl
4    Department of Electrical Engineering, Faculty of Science and Technology, Mostaganem University,
     Mostaganem 27000, Algeria; mohammed.guezgouz@univ-mosta.dz
*    Correspondence: jakubkamiljurasz@gmail.com

**Abstract:** In recent years, Poland has experienced a significant increase in the installed capacity of solar and wind power plants. Renewables are gaining increasing interest not only because of Poland's obligations to European Union policies, but also because they are becoming cheaper. Wind and solar energy are fairly-well investigated technologies in Poland and new reports are quite frequently added to the existing research works documenting their potential and the issues related to their use. In this article, we analyze the spatial and temporal behavior of solar and wind resources based on reanalysis datasets from ERA5. This reanalysis has been selected because it has appropriate spatial and temporal resolution and fits the field measurements well. The presented analysis focuses only on the availability of energy potential/resources, so characteristics intrinsic to energy conversion (like wind turbine power curve) were not considered. The analysis considered the last 40 years (1980–2019) of available data. The Spearman coefficient of correlation was considered as a complementarity metric, and the Mann–Kendal test was used to assess the statistical significance of trends. The results revealed that: The temporal complementarity between solar and wind resources exists mostly on a seasonal scale and is almost negligible for daily and hourly observations. Moreover, solar and wind resources in joint operation exhibit a smoother availability pattern (assessed based on coefficient of variation). Further findings show that the probability of 'resource droughts' (periods when cumulative generation was less than arbitrary threshold) lasting one day is 11.5% for solar resources, 21.3% for wind resources and only 6.2% if both resources are considered in a joint resource evaluation. This situation strongly favors the growth of local hybrid systems, as their combined power output would exhibit lower variability and intermittency, thus decreasing storage demand and/or smoothing power system operation.

**Keywords:** solar energy; wind energy; complementarity map; theoretical availability; seasonal complementarity

## 1. Introduction

Renewable energy sources play various roles across the globe [1]. The uneven distribution [2] of their installed capacity is a matter, not only of local availability of renewable resources, but also of national policy and available capital.

As the share of renewables grows, researchers are beginning to investigate their characteristics in more depth [3]. Of particular interest is their spatial and temporal behavior, as these qualities heavily impact the performance of energy systems [4]—both large-scale, interconnected ones as in Europe [5] and small-scale, off-grid systems supplying power to local communities or, for example, water pumping devices [6].

The two recently most intensively expanding renewable energy sources, namely wind and solar, are known for their intrinsically weather- and climate-driven nature. Not only does their availability often not match demand, but they also increase variability on the supply side. This situation calls for greater power-system flexibility, which can be achieved in many ways by, for example, the integration of energy systems (sector coupling) [7] as well as methods like fuel shifting [8], energy storage [9], demand response [10] or hybrid energy systems, based on complementarity of renewable resources [11,12]. For instance, Sun et al. [13] explored how the complementarity of grid-connected hybrid solar–wind systems could improve the exploitation of these resources. Han et al. [14] developed a methodology to evaluate the complementarity between solar, wind and hydropower in terms of power fluctuations and ramp. Their analyses confirmed the effectiveness of the proposed method against Kendall's rank coefficient of correlation. Kapica [15] analyzed solar- and wind-energy availability to find the relationship between installed capacity and storage required to achieve a supply reliability of 95% for a small-scale system in Europe. Neto et al. [16] used a general approach based on Daily Physical Guarantee to assess the impact of complementarity on the operation of solar, wind and tidal in terms of a microgrid system. Their results indicated that the diversification of anti-correlated renewables supports the decarbonization of an isolated microgrid system.

The complementarity concept was not only studied for solar and wind energy resources but also for other technologies such as a combined heat and power station (CHP), run-of-the-river (RoR) and large-scale hydropower. Puspitarini et al. [17] investigated the combination of solar photovoltaic panels, run-of-the-river and a combined heat and power station in a 100% renewable energy mix scenario. This study was conducted for seventeen districts in north-eastern Italy, where its results indicated that integration of the CHP system is a promising solution to strengthen the energy balance and increase load satisfaction. Puspitarini et al. [18] explicitly studied the effect of rising temperatures and glacier melt on electricity production and the solar–RoR energetic complementarity in the eastern Italian Alps. Complementarity was assessed based on three parameters, namely, Pearson coefficient of correlation, standard deviation and demand satisfaction. Their findings showed that the increase in temperature sped up the process of ice melt, which implies an increase in power generation during summer. Therefore, the authors recommend it, including glacier changes in the future estimation of complementarity between solar and RoR power since it is found that glacier shrinkage has different impacts on the complementarity.

Moreover, the complementarity of solar–hydro is proposed in several studies, for different purposes such as covering residual load and long-term scheduling operation of hydropower. For example, Rauf et al. [19] proposed the utilization of floating solar PV (FSPV) in addition to hydroelectricity to cope with the daytime electricity peak demand. Their results suggested that the complementing effect that characterizes FSPV and hydropower saves water for late-night hours and gives the grid operator more flexibility and stability. Danso et al. [20] used a dynamic programming algorithm to examine the contribution of a large hydropower reservoir to cope with the residual load for various shares of solar and wind energy. Their results revealed that hydropower could cover the residual load in the case that solar and wind contributed less than 20% to meeting overall energy demand. In the case of high solar penetration, the residual load could not be fully smoothed, thus, requiring the use of a short-term storage system. Zhu et al. [21] established a framework based on nonlinear modeling and multi-objective optimization for long-term complementarity operation of the Longyangxia hydro–PV plan in China. For the same location, Li et al. [22] applied dynamic stochastic programming to optimize the long-term complementarity operation of this hydro–PV plant. Both these last two studies stress the importance of the complementarity concept in planning the operation of a hybrid hydro–PV station.

Four recently published review articles [23–26] have archived, analyzed and discussed the concept of renewable energy source complementarity. The reviews indicate that, in

recent years, significant effort has been dedicated to understanding, quantifying, visualizing and interpreting the meaning of the complementarity concept. It has been indicated that complementarity should be referred to not only as a parameter that can be calculated but also as a measure or property that can be modified (for example, by adjusting the orientation of PV modules in PV–wind systems) or used in the context of a whole hybrid system's operation. The latter most cases involves hybridization in terms of the operation of solar/wind and hydropower stations, as discussed extensively in many works [27–30].

The aforementioned review articles have highlighted the existing gaps in knowledge about the complementarity of renewable energy resources. One such gap is the lack of complete global spatial coverage of complementarity studies. In many countries, no studies have been conducted, and in others, often, few locations have been selected. This work aims to shed some light on renewables' complementarity in Poland, as well as their seasonal variability. Additionally, the objective of this work is to investigate the potential occurrence of 'energy droughts' (hereafter 'resource droughts), which has been defined in [31] as periods when energy generation from renewables was bellow certain threshold. A few papers have so far been published on this issue for Poland (as indicated in [23]), but their spatial coverage is not explicit. Summarizing the above, this research work aims to answer the following questions:

- What is the spatial and temporal nature of solar and wind resource availability in Poland?
- How does the temporal complementarity between solar and wind resources vary over the area of Poland?
- What is the probability of 'resource droughts' occurring in different regions, and how is it affected by the hybridization of solar and wind resources?

The remainder of this paper is organized as follows: Section 2 briefly presents the data used and methods applied. Section 3 reports and discusses the major study findings. The article ends with Section 4, which summarizes all results, provides final conclusions and highlights further potential research directions.

## 2. Materials and Methods

Assessing complementarity of solar and wind energy requires long-term meteorological data representative for the considered region with high spatial resolution. On the other hand, numerous methods have been proposed in literature to quantify the complementarity between the underlying resources such as correlation, reliability indexes, output fluctuation and other relevant methods [23]. In this regard, the following section firstly discusses the data and secondly the methods applied in our analysis.

### 2.1. Input Data

For the analysis, hourly-resolution data on wind speed at 100 m above ground level (a.g.l.) and solar irradiation on horizontal surface has been downloaded for the area between E: 14.00°–24.00° and N: 48.75°–55.50° [32] covering the years 1980–2019. The data from ERA5 (a fifth-generation reanalysis by the European Center for Medium-Range Weather Forecasts—ECMWF, with hourly data on single levels) comes with a fine spatial resolution ($0.25° \times 0.25°$) and several studies have shown it to precisely fit with ground measurements, including for Poland [33], or have found its usage satisfactory for energy system modeling [34,35].

The solar energy resources time series for analysis were taken directly from ERA5, where they are available as shortwave radiation reaching the horizontal surface. Therefore, no further processing was applied.

In the case of wind energy potential, the wind speed from ERA5 is available at 100 m a.g.l., which, conveniently, is also the typical hub height of modern wind turbines. However, wind speed is provided as $u$ (eastward component) and $v$ (northward component), which have to be combined to yield the speed and direction of the horizontal 100-m wind.

The power (Equation (1)) that can be intercepted by the blades of a wind turbine is a function of air density, area swept by the blades and the cube of the wind velocity (if efficiencies are neglected) [36,37]. The calculations take no account of, either temporal or spatial variation of air density, and a value of 1.225 kg/m$^3$ was assumed,

$$P(t) = \frac{1}{2}\rho_{air} A v^3, \tag{1}$$

where: $P(t)$—power generated at time $t$ (W), $\rho_{air}$—air density (kg/m$^3$), $A$—area swept by the rotor blades (m$^2$), $v$—wind speed (m/s).

In the article, we have purposefully refrained from converting energy potential to the energy generation from respective energy power plants, i.e., PV modules and wind turbines. The work objective is to focus on resources, and not on the technical solutions used to obtain energy. Since many works have reported ways to increase the efficient utilization of renewables, like suboptimal PV module orientation, which decreases energy yield but improves supply/demand fit [38], or optimizing PV orientation with regard to small-scale hydro in order to increase complementarity [39–41]. This approach saves us from limiting the results to arbitrarily selected PV modules and wind turbines (with their intrinsic energy conversion efficiencies), but also opens promising future research directions, as discussed later in the article conclusions.

### 2.2. Temporal Complementarity

For the temporal complementarity ('understood as a phenomenon when variable renewable energy sources' exhibit periods of availability which are complementary in the time domain' [23]), an index based on the Spearman coefficient of correlation (Equation (2)) was selected, as correlation coefficients are metrics commonly applied in the literature [23,25]. The major benefits of this metric are its straightforward calculation and easy interpretation. Since the values of Spearman coefficient of correlation range from $-1$ to 1, those closer to $-1$ are associated with the beneficial complementary nature of two energy sources, whereas values closer to 1 indicate that the analyzed energy sources will likely follow the same energy generation pattern in time,

$$\rho = 1 - \frac{6 \sum d_i^2}{n(n^2 - 1)}, \tag{2}$$

where: $\rho$—Spearman rank correlation, $d_i$—difference between ranks of corresponding variables, $n$—number of observations.

The typical interpretation of the correlation coefficient as a metric of the complementarity of two renewable energy sources (*Source_A*, *Source_B*) is visualized in Figure 1. The data prepared for this figure has been modeled based on sine waves and the complementarity was modified by changing the phase angle ($\varphi$) as shown in Equations (3) and (4):

$$Source\_A = \sin(t), \tag{3}$$

$$Source\_B = \sin(t + \varphi). \tag{4}$$

Figure 1 presents four instances of two potentially complementary energy sources. Their generation time series are idealized, meaning that, for clarity, they are simulated based on sine functions. The complementarity is expressed as the Pearson coefficient (since the time-series meet the assumptions required by this index) of correlation and marked as $\rho$ in the titles. The top-left chart presents the situation in which two resources are perfectly correlated. This might be the case of, for example, two PV installations located in close proximity (as indicated in [23], complementarity can also be assessed between the same energy sources). In such a situation, the energy availability over time will not change, regardless of the change in capacity of power plants—meaning there will be periods of high generation and of very low or zero generation (PV in the night). The top-right and

bottom-left figures present situations of a slow tendency for the synergetic/complementary effect of the two power plants' operation to appear. The analysis of those charts makes it clear that the occurrence of extreme low and extreme high generation periods decreases and there is also a better match with the demand. The ideal situation is presented in the bottom-right chart. There, the complementarity between the two resources is perfect, meaning that they ideally complement each other's generation pattern, and the energy demand (black line) is always satisfied.

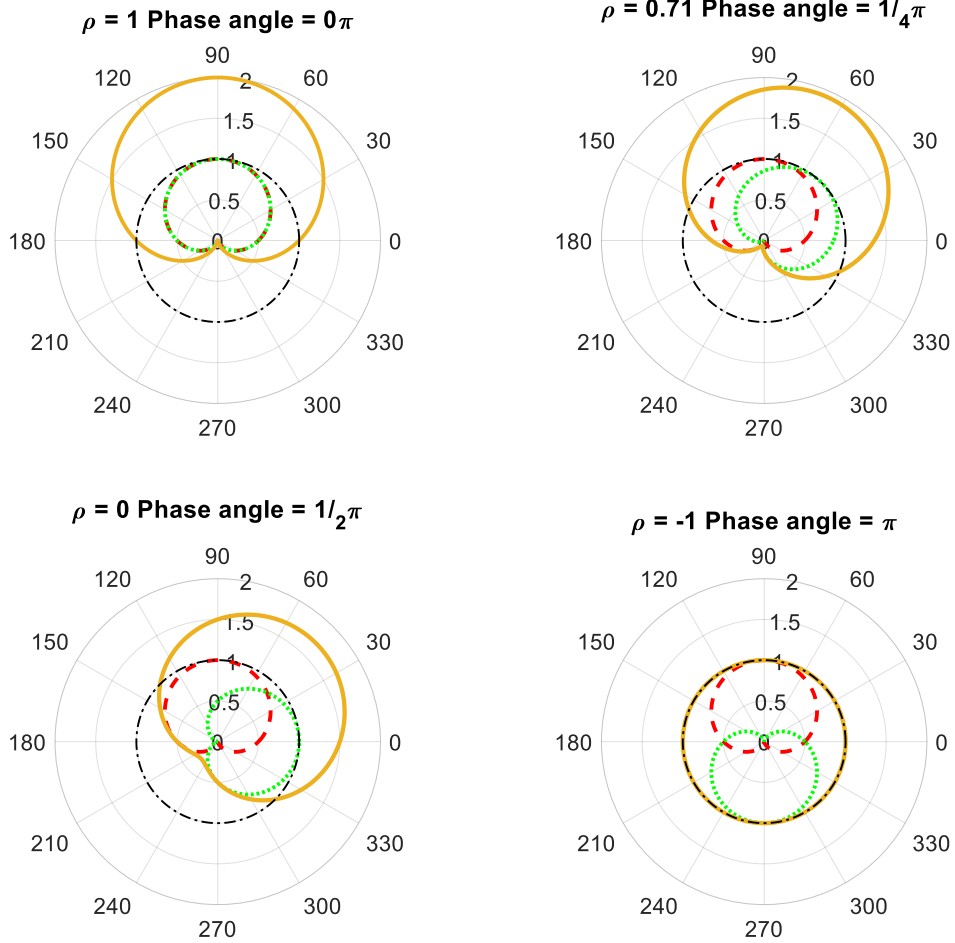

**Figure 1.** Representation of four different scenarios in which two renewable energy power plants (red and greed dotted and dashed lines, respectively) exhibit different complementarity (here expressed as Pearson coefficient of correlation—$\rho$). The phase angle represents the angular shift between two sine functions. The continuous (orange) line represents the joint energy generation from the two energy sources, whereas the dot-dash (black) line indicates a theoretical constant load. If the orange line is above, it indicates an energy generation surplus, and when it is below, energy demand is greater than supply, and energy deficits occur.

### 2.3. Spatial Smoothing/Decorealation

It is a well-established fact that spatially distributed, variable renewable energy sources tend to smooth their aggregated availability time series [42]. It is observed that with increasing distance between sites, the correlation between their energy generation time series decreases. Here again, a Spearman coefficient of correlation can be applied, and the distance between sites can be calculated based on the *haversine* formula. More specifically, a correlation matrix has been created which contained information about the correlation coefficient between hourly resource availability between all locations. For example, if 10 locations were considered in total 45 values of the Spearman coefficient of correlation were considered (the entries above or below the main diagonal are relevant). In the same

way, the distance matrix has been developed. Finally, the figures visualizing the spatial smoothing/decorrelation has been made by plotting correlation against the distance.

### 2.4. 'Resource Droughts'

The first objective of this work is to analyze the complementarity of wind and solar resources over the land areas of Poland. The second focuses on examining the spatial smoothing effect. The third part focuses on potential extreme events relating to the availability of solar and wind energy. Here we have focused on extreme low generation periods, also called 'resource droughts'. We have adopted a definition proposed recently by [31] that defines 'energy production droughts' (which we here refer to instead as 'resource droughts') as periods when daily resource availability is 0.2 times less than the mean daily availability ($T$) over the considered period. Thus, a daily time step is considered in the analysis. The concept of 'resource drought' can be expressed as in Equation (5),

$$RD_i = \begin{cases} 1 \; if \; EA_i \leq T \\ 0 \; otherwise \end{cases}, \tag{5}$$

where: $RD_i$—'resource drought' index (binary), $RA_i$—resource availability (kWh) expressed as the daily sum of available solar and wind energy, and $T$—daily energy availability threshold, set here as 0.2 of mean daily production/availability over the considered period.

As a consequence, we see that the 'resource drought' index becomes a binary time series in which "0" indicates days with generation over the threshold and "1" the opposite. Having such time series is a straightforward way to calculate the duration and frequency of 'resource drought' periods. Here, we consider solar and wind resources both independently and in combination in order to investigate how they complement each other and impact the occurrence of energy droughts.

### 2.5. Statistical Significance of Trends

There are several statistical tests for detecting trends in meteorological time series (e.g., linear regression test and the Mann–Kendall test). This research employs the non-parametric Mann–Kendall test widely applied in meteorological studies [43–45]. This test uses rank correlation statistics to answer the question of whether the values measured in the time series present a statistically significant increasing or decreasing trend. The Mann–Kendall test analyses the sign of the difference between successive elements of the time series, given as $x_1 \leq x_2 \leq \ldots \leq x_t$, where each new value is compared to all previously measured values [46]. The following equation calculates the statistic $S$,

$$S = \sum_{i=1}^{n-1} \sum_{j=i+1}^{n} sign(x_j - x_i) \tag{6}$$

where the indexes $j$ and $i$ emphasize all possible pairs composed of earlier and later values of a deseasonalized (e.g., annual) series. By substituting $x_j - x_i = \theta$ we get:

$$sign(\theta) = \begin{cases} +1 \; for \; \theta > 0 \\ 0 \; for \; \theta = 0 \\ -1 \; for \; \theta < 0 \end{cases} \tag{7}$$

where $sign(\theta)$ is the positive or negative sign of the parenthetical result. If the statistic $S$ is positive, the more recent measurements are higher than the earlier ones, indicating an upward trend in the measured values. If $S$ is negative, there is a downward trend. The

rate of change in the analysed trend can be described by the directional coefficient of the straight line expressed by Sen's slope estimator $\beta$:

$$\beta = median\left(\frac{x_j - x_i}{j - i}\right) \tag{8}$$

calculated for every $i < j$, where $i = 1, 2, \ldots, n - 1$ and $j = 2, 3, \ldots, n$.

According to [47], the null hypothesis is the absence of a trend ($H_0 : \beta = 0$) and the alternative hypothesis is that a trend is detected ($H_1 : \beta \neq 0$) that is statistically significant above the 95% confidence level.

As a final remark, input datasets have been obtained as netCDF files, processed in Matlab software and, finally, visualized in both Matlab and ArcGIS.

## 3. Results and Discussion

This section presents the results of the conducted analysis and discusses them in the light of the available literature. The section is divided into several subsections, starting with a general analysis of spatial and temporal variability of solar and wind energy. The second section discusses the complementarity of these two energy sources and the last section is dedicated to the phenomenon of 'resource droughts'.

### 3.1. Spatial and Temporal Variability of Solar and Wind Energy

Numerous past works [48–52] have been dedicated to the analysis of renewable energy sources potential in Poland. However, relatively little attention has been paid to the potential of both solar and wind resources from the perspective of the whole country and their complementary nature. In this article the ERA5 re-analysis data has been applied and revealed known spatial features in terms of solar and wind resources. For wind energy (Figure 2), it was found that the highest potential is observed close to the Baltic Sea coastal zone. This is a well-known situation confirmed both by the prevailing capacity of wind power plants installed in the northern voivodeships (NUTS2-level administrative divisions) and by energy companies' interest in developing off-shore wind parks in the Polish part of the Baltic Sea. When it comes to solar energy (Figure 3), visualization confirms a higher potential in southern Poland. The annual sums are also in line with other data sources like www.solargis.info and range around 1000–1200 kWh/m$^2$. In previous work [33], we ran a comparison between ground measurements and data from sources like ERA5, CAMS and MERRA, and found that the ERA5 reanalysis has a good match with measurements. This evidences its usefulness in energy-sector-related analysis.

In the case of solar and wind resources, we have observed statistically significant trends indicating that the potential of both energy sources has changed over the last 40 years. In the case of wind resources, downward trends were observed in central Poland (52.00° S, 19.00° E), as opposed to upward trends for solar over the whole of Poland. The spatial representation of trends (Figures 2 and 3) is additionally supported by national average resources availability, as presented in Figures 4 and 5. Those figures confirm not only the observed trends but also a relatively high inter-annual variability of resources. The highest differences between individual years are especially visible for wind resources, which in 1996 slightly exceeded 1800 kWh/m$^2$ at 100 m a.g.l., but in 1983 were over 50% greater, amounting to 2813 kWh/m$^2$. Such a huge inter-annual variability of solar and—especially—wind resources indicates that decision makers should consider the fact that there might be years when energy generation from those sources will be significantly below average and alternative energy sources would have to be considered.

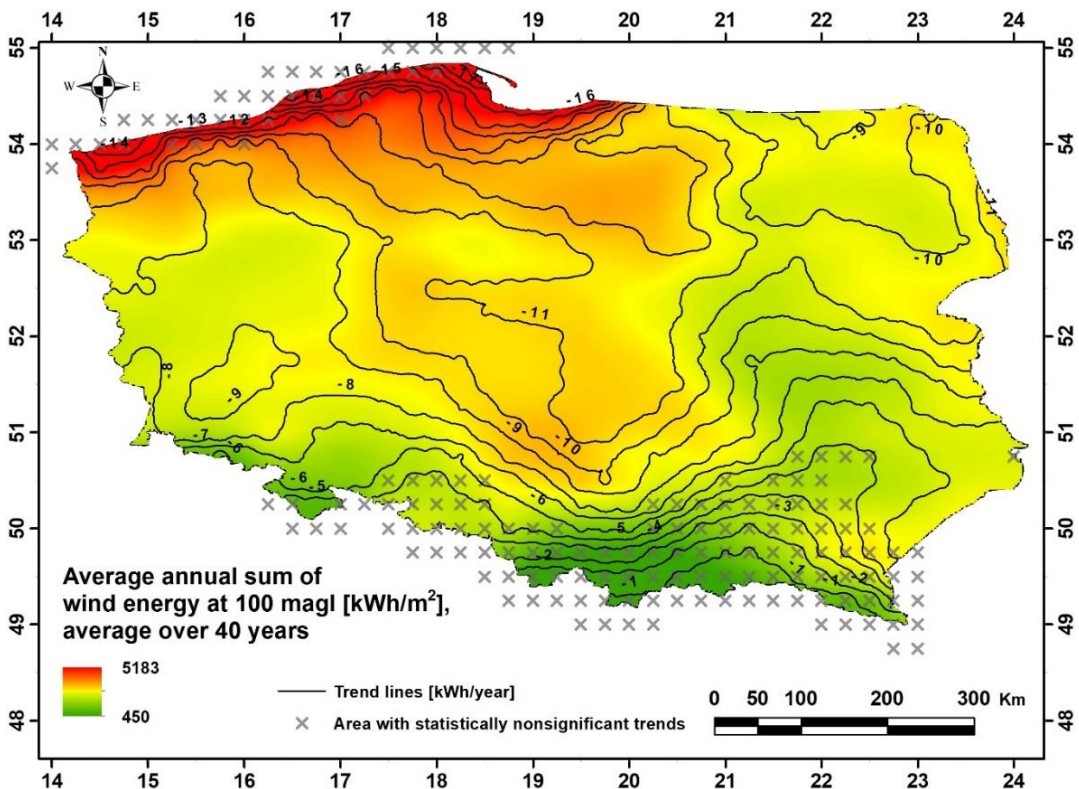

**Figure 2.** Spatial distribution of wind resources in Poland. Data represents mean resource availability at 100 m a.g.l. calculated from hourly time series for 1980–2019. Contour lines represent linear trends.

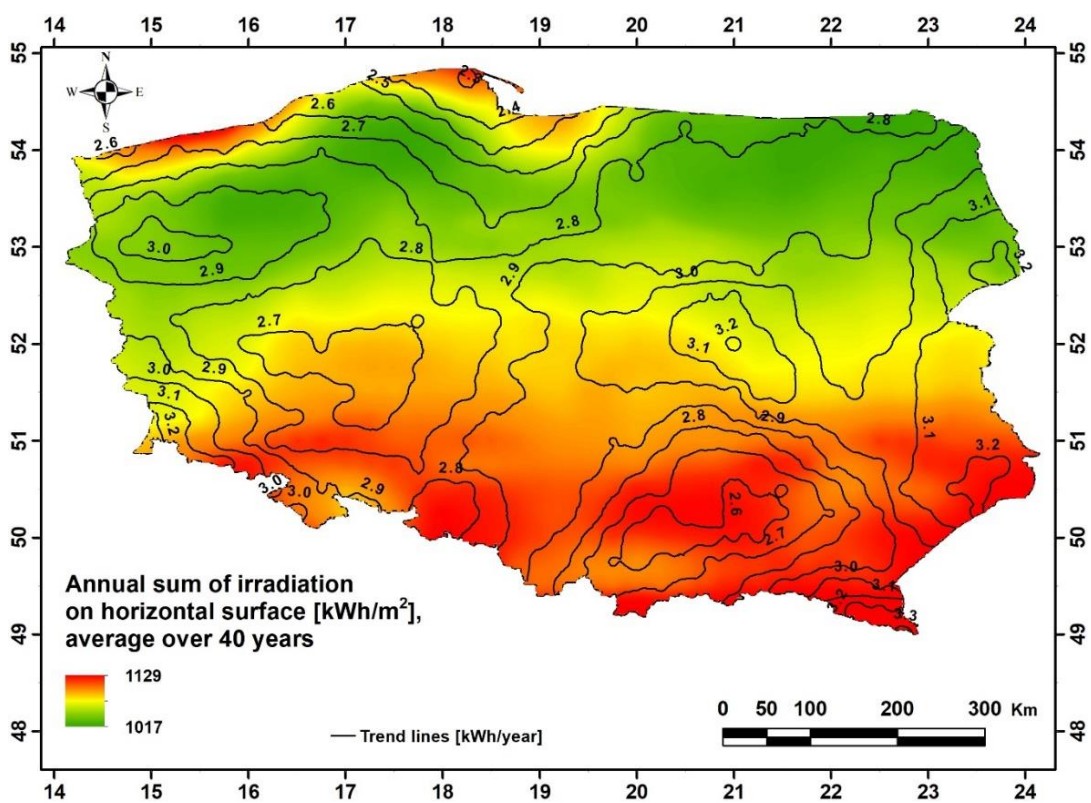

**Figure 3.** Spatial distribution of solar resources in Poland. Data represents mean resource availability on horizontal surface calculated from hourly time series for 1980–2019. Contour lines represent linear trends.

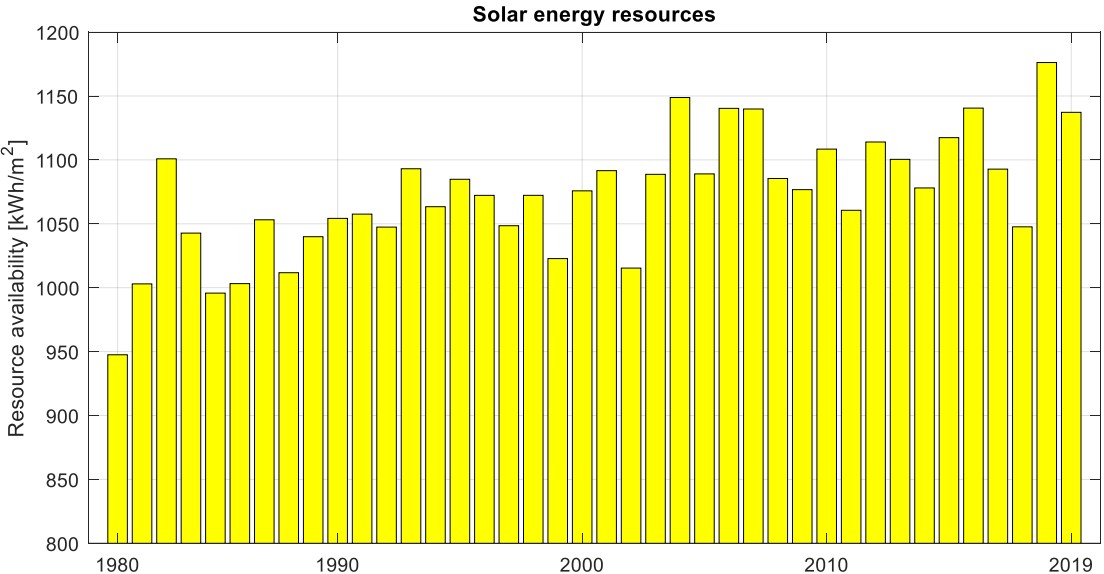

**Figure 4.** Multi-annual (1980–2019) change in country-wide mean annual sum of irradiation on horizontal surface. Calculations considered only ERA5 grid points within Polish borders.

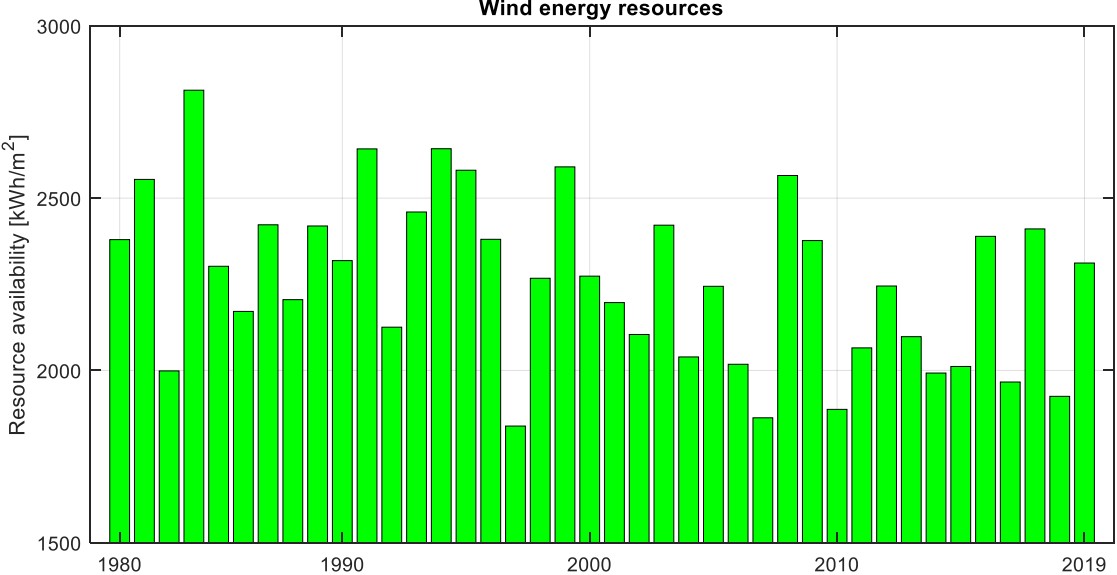

**Figure 5.** Multi-annual (1980–2019) change in country-wide mean annual sum of wind energy at 100 m a.g.l. Calculations considered only ERA5 grid points within Polish borders.

Our findings with regard to wind resources potential are in line with results obtained by Jung et al., [53] who have investigated the long-term trends of wind potential across the globe. In particular they found for Poland relative change of decadal wind energy generation to be a maximum of −2.49%. Although their approach is different than ours (power curves vs. energy potential) the results are consistent. Importantly their follow up work [54], based on climate projection models (focusing on Germany—also Central Europe as Poland) highlights the declining capacity factors of wind generators in the upcoming centuries.

In terms of solar energy, Pfeifroth et al., [55] analysed the variability and trends of surface solar radiation based on CMSAF's SARAH-2 and CLARA-A2 climate data records in Europe. While the authors used different datasets than the used herein, their findings are in agreement with ours and it showed positive trends in solar radiation changes from 1983 to 2015. Sanchez-Lorenzo et al., [56] studied the changes in solar radiation over Europe

using The Satellite Application Facility for Climate Monitoring (CM SAF) datasets from 1983 to 2010. The method used to assess the trends was almost similar to the one applied here in our work. Additionally, their outcomes are in concordances with our results. Where it indicated an increase of solar radiation up to an average of 2 W/m$^2$ per decade. In summary, the previously mentioned studies confirm the consistency of the upward trends of solar resources in Europe and particularly in Poland.

Here, an additional question emerges, namely, whether solar and wind energy can complement each other on an inter-annual timescale? In the earlier sections we observed that the trends clearly indicate that diminishing wind resources are to some extent replaced by solar energy. However, a further analysis is needed to quantify this replacement and its impact on long-term power system capacity planning. In Figure 6, we present the complementarity between solar and wind resources from the inter-annual perspective. It clearly shows that in the southern region (mostly mountains and highlands—below the irregular yellow line linking 51.75° N on the left vertical axis to 49.50° N on the right axis) there is a lack of significant negative correlation between these two energy resources. However, in the rest of Poland (apart from some regions on the northern border) we observe a negative correlation between solar and wind resources. This indicates that in these regions the years with lower availability of wind resources will to some extent be compensated by higher solar energy availability, and vice versa. To support our claim that some inter-annual complementarity exists between solar and wind energy, Figure 7 represents, as a scatter plot, the relationship between solar and wind resources across all locations over the period of 40 years. After fitting the linear trend, we found it to be negative and statistically significant.

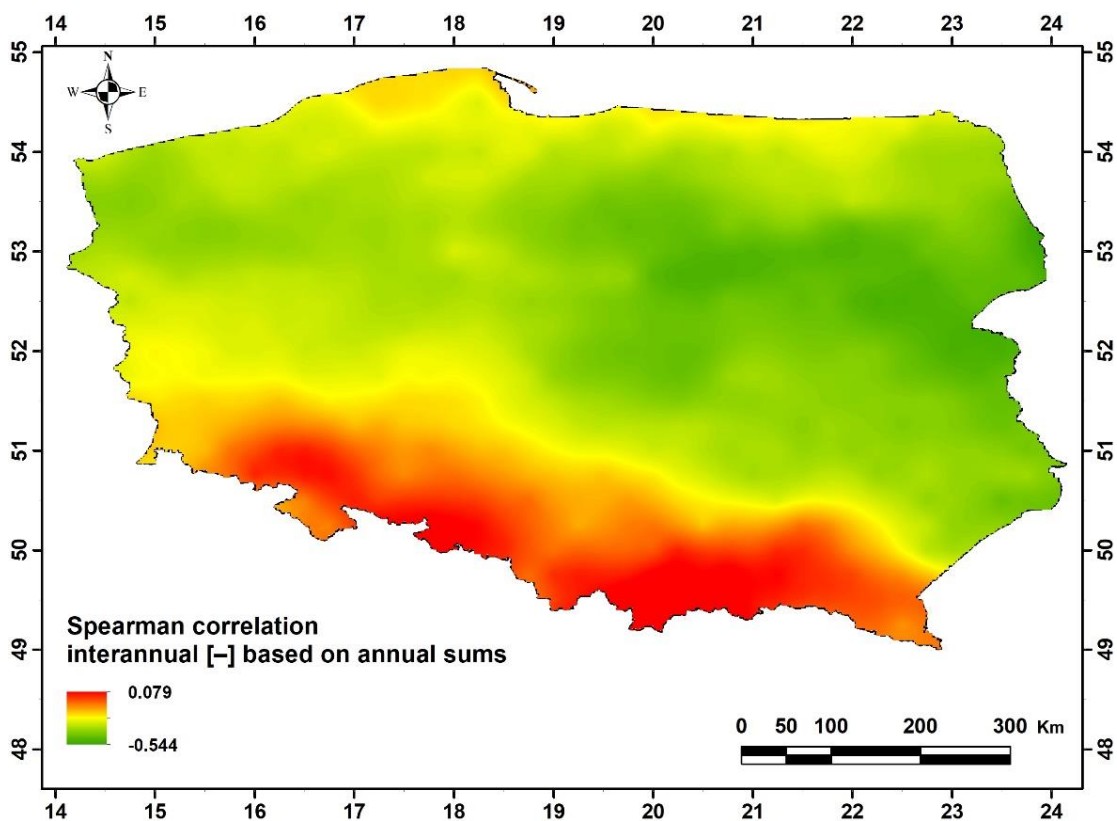

**Figure 6.** Interannual correlation between solar and wind resources across Poland, based on annual sums of resources availability, 1980–2019.

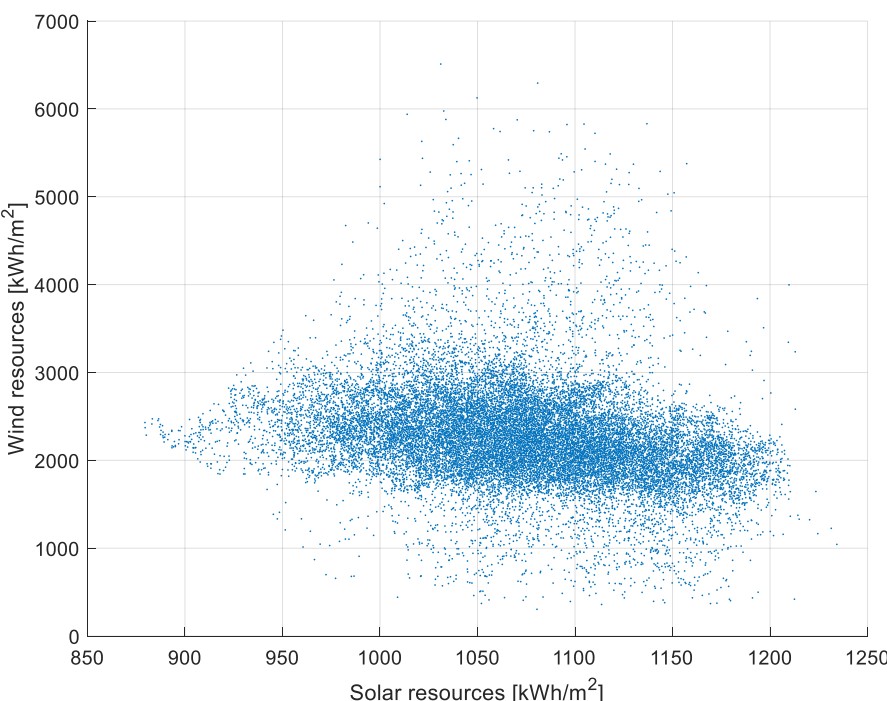

**Figure 7.** Inter-annual relationship between solar and wind resources for all grid points within the borders of Poland, 1980–2019.

### 3.2. Temporal Complementarity of Solar and Wind Resources

The previous section discussed the spatial distribution of solar and wind resources across Poland during the years 1980–2019 and their inter-annual availability and complementarity. Here, we present and analyze the existence of any temporal complementarity between solar and wind energy within the current borders of Poland. The areas considered for potential off-shore wind farms are not taken into consideration. The temporal complementarity has been expressed in terms of Spearman coefficient of correlation as presented in Section 2.2 and analyzed for three-time scales, namely, monthly sums, daily sums and hourly values of resources availability.

The assessment results have been combined and visualized in Figures 8 and 9. As can be observed, the temporal complementarity exhibits a significant spatial variation in Poland. The southern regions (mountains) are characterized by significantly lower complementarity (indicated by the lower value of the Spearman coefficient of correlation). For the hourly time step, there is almost no complementarity to speak of between solar and wind resources, since the Spearman coefficient of correlation is, on average, close to 0 (precisely, −0.16). This result indicates that on an hourly time scale it is highly unlikely that a sudden decrease in energy generation from one system will be compensated by the other. The situation improves slightly for complementarity assessed in terms of daily sums. The Spearman coefficient of correlation is on average −0.30, which is still a weak correlation, but it might suggest that, from time to time, days with low generation from renewable sources like solar energy will be supported by higher generation from wind sources. The highest daily complementarity index values (−0.37) were observed for south-eastern Poland (23.50° E, 49.50° N). The low potential complementarity on both daily and hourly time scale clearly indicates the high need for short-term energy storage solutions, in particular, in the case of off-grid energy systems, as shown in example, [57].

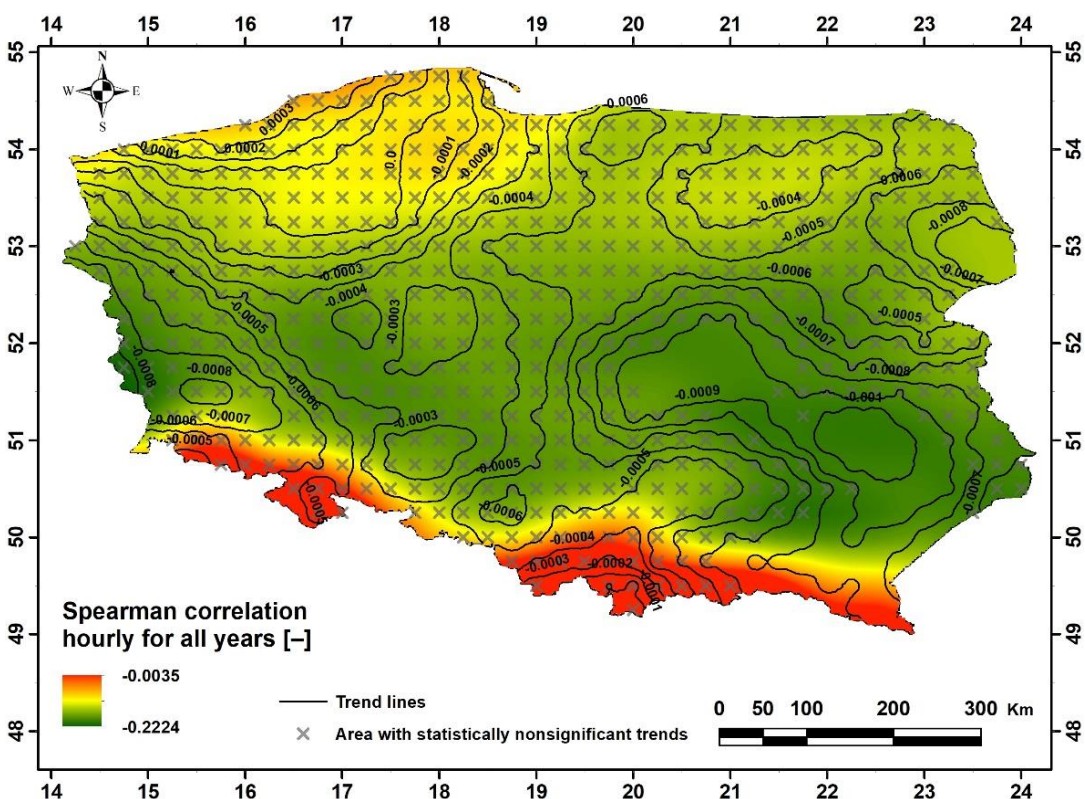

**Figure 8.** Complementarity between solar and wind resources based on hourly resources availability, 1980–2019. Crosses show where trends (isolines) are not statistically significant.

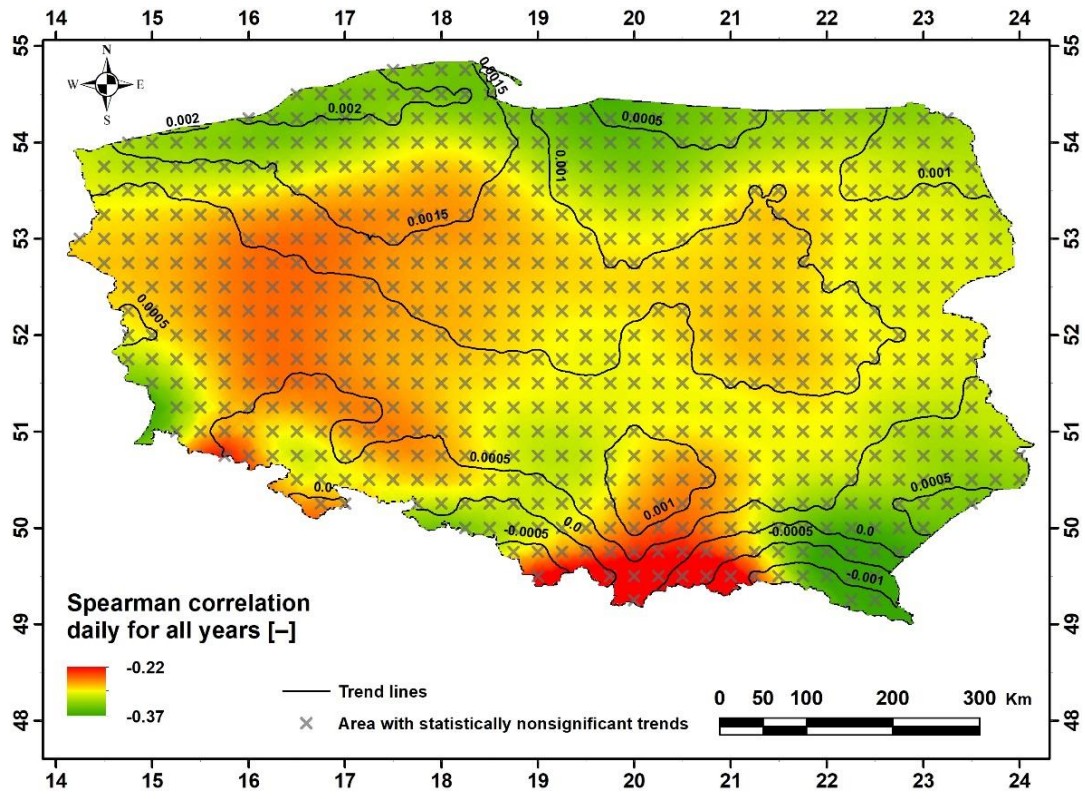

**Figure 9.** Complementarity between solar and wind resources based on daily sums of resources availability, 1980–2019. Crosses show where trends (isolines) are not statistically significant.

The complementarity assessed based on monthly sums of solar and wind energy resources availability is quite variable. The highest values (in terms of complementarity) are observed, again, in Poland's most south-easterly province of Subcarpathia. Whereas, the lowest ones (positive values of the Spearman coefficient of correlation) are found in its immediate western neighbor of Lesser Poland (and in particular in the Tatra Mountains range). The observed monthly complementarity between solar and wind resources over certain areas of Poland indicates that over the course of the year a seasonal synergetic effect between solar and wind resources might be observed. The full analysis of the complementarity of solar and wind resources in Poland across the three proposed time steps indicates, but that energy storage is required if reliable power supply to load is to be maintained, in particular for off-grid systems.

Apart from the pure analysis of the complementarity between solar and wind resources, we have enhanced our analysis by investigating whether there are any significant trends over the considered period. On the maps presented in Figures 8–10, apart from complementarity we have also presented trends (slope is given on isolines) and crosses indicate that the trend was not statistically significant (Section 2.4). The presented results make it clear that although some trends exist they are statistically significant in only a few regions (in particular for hourly complementarity).

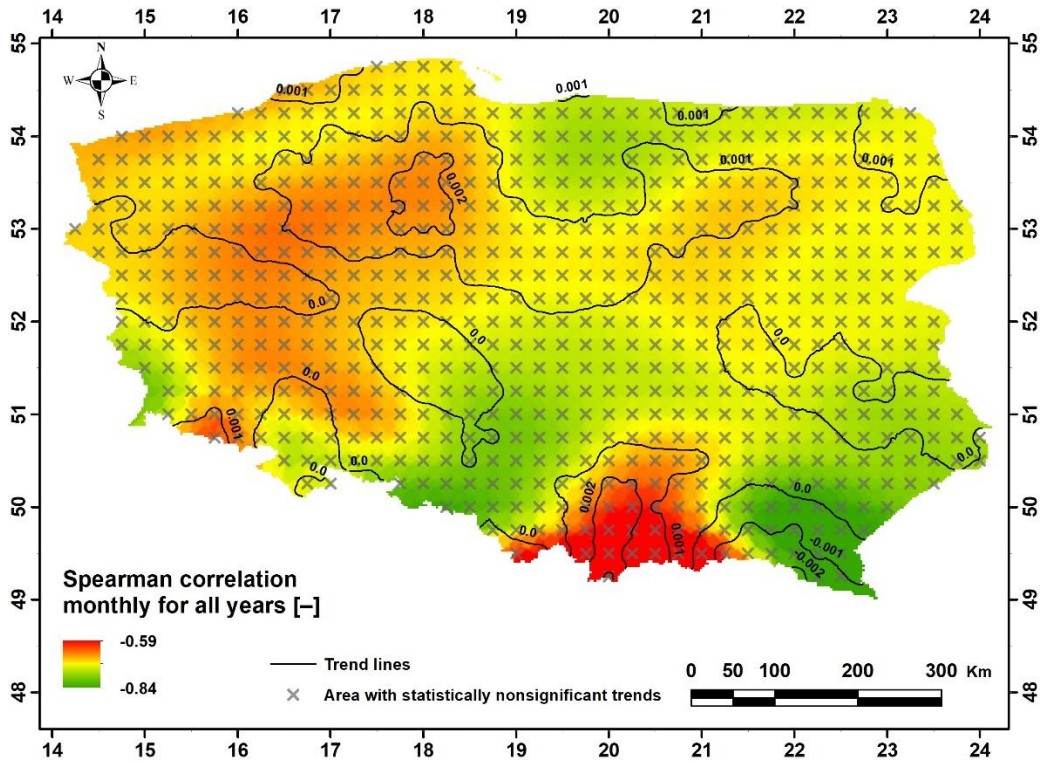

**Figure 10.** Complementarity between solar and wind resources based on monthly sums of resources availability, 1980–2019. Crosses show where trends (isolines) were not statistically significant.

We have already shown in earlier paragraphs that there exists a temporal complementarity between solar and wind energy resources in Poland, but that it tends to vary a lot depending on time step. It was found that, for the hourly time step, the average coefficient of correlation (selected here as a metric for complementarity evaluation) ranges around −0.16. The question, thus, arises on whether hybridization of solar and wind technology raises or lowers the variability of energy output of an energy source. The variability of hourly resources is illustrated using the coefficient of variation (defined as standard deviation divided by mean value) for solar (Figure 11), wind (Figure 12) and combined solar–wind (Figure 13). The lower the value, the more stable the power generation from the

energy source. The coefficient of variation of the national load can be used as a reference, and was 16.3% for Poland for 2018 (based on data available at the Transmission System Operator website).

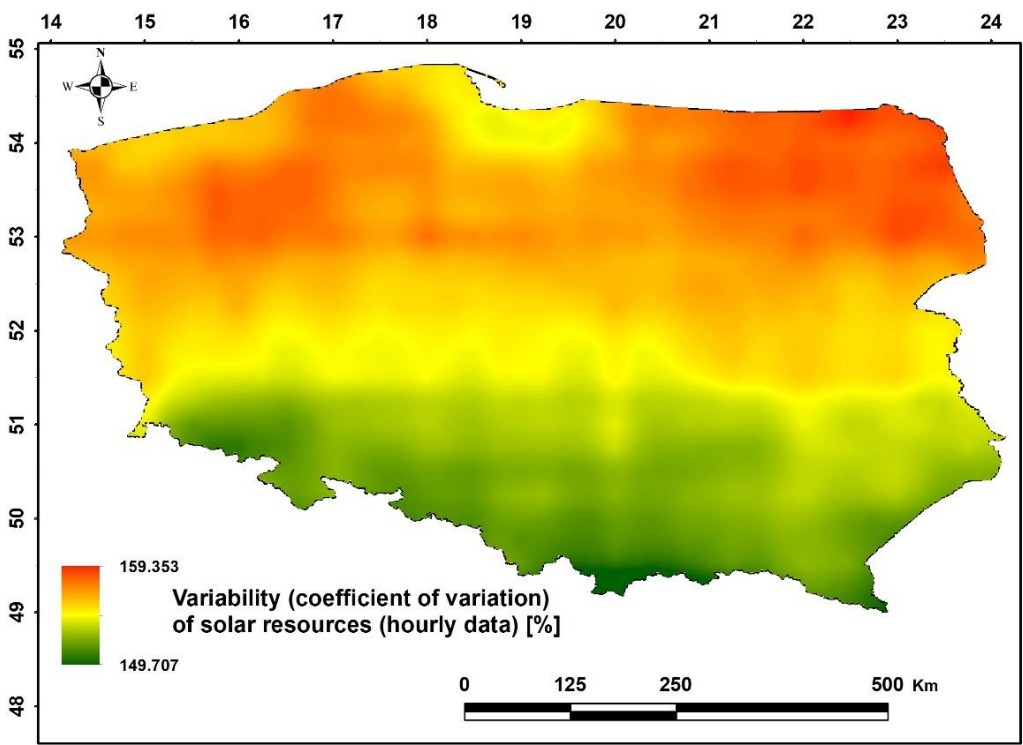

**Figure 11.** Solar resources coefficient of variation, 1980–2019.

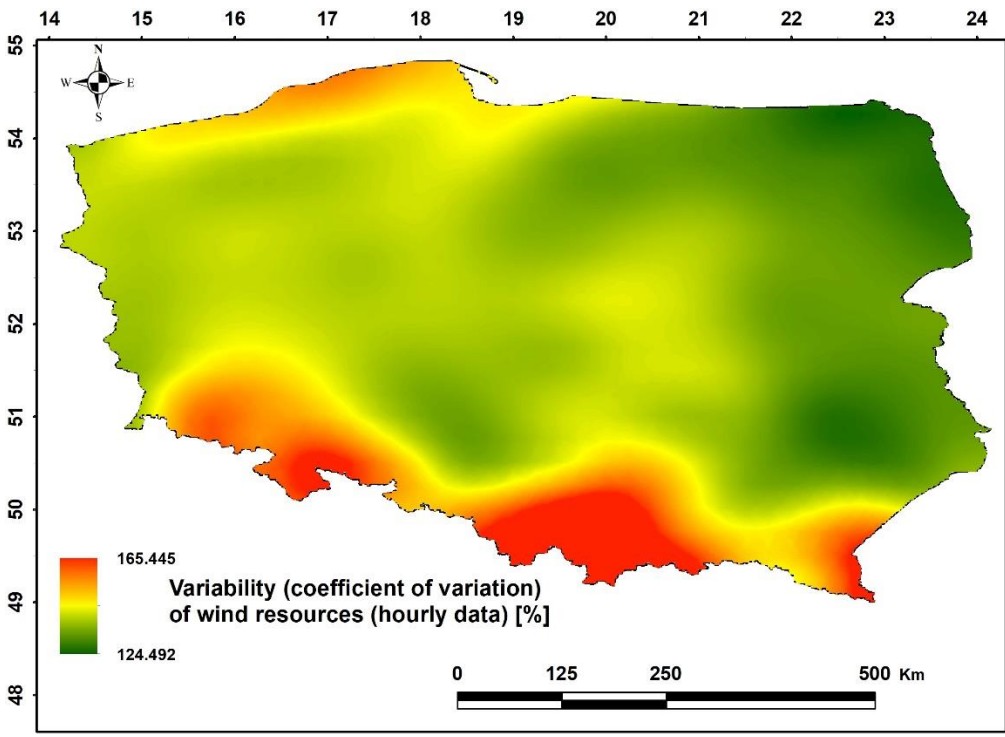

**Figure 12.** Wind resources coefficient of variation, 1980–2019.

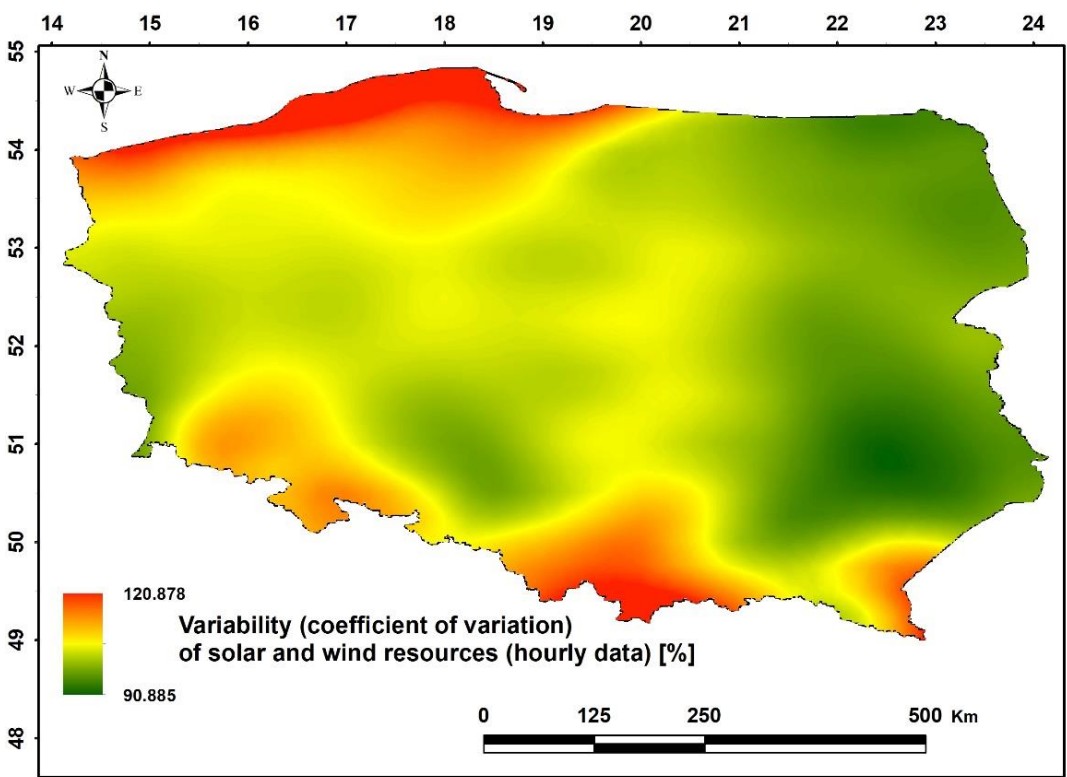

**Figure 13.** Solar+wind resources coefficient of variation, 1980–2019.

The coefficients of variations for solar and wind resources are much higher than demand, and are on average 155.3% and 139.1%, respectively. Furthermore, there is also a spatial variability of this parameter. Solar resources were found to vary most in north-eastern Poland (54.00° N, 23.00° E), and wind resources in southern Poland (in particular Subcarpathia, Lesser Poland, Lower Silesia). The highest recorded values were 160%, and 197.5% for solar and wind resources, respectively. However, when the solar and wind resources are investigated jointly (the hybrid system) the coefficient of variation drops. For the whole of Poland, it was found to average 101.1%. The highest values are again observed in southern Poland, but these regions are simultaneously characterized by relatively low wind energy potential. Nevertheless, it must be noted that in particular locations in southern Poland, too, wind parks have successfully been installed. The drop in the coefficient of variation indicates that a hybridized energy source, namely one that simultaneously utilizes both solar and wind resources, will have a smoother and less variable power output. Such sites can be found especially in north-western Poland. This leads us directly to the next section, which is dedicated to the effect of spatial smoothing of variable renewable energy sources.

### 3.3. Spatial Smoothing/Decorrelation

Spatial smoothing is the phenomenon of the joint generation curve of separate power plants being smoothed as distance increases between the power plants. One way of testing this phenomenon is by comparing the variability of joint and separate generation time series. Another approach is based on correlation coefficient and the decorrelation effect, i.e., the tendency for correlation to decrease with increasing distance between sites. For Polish wind and solar resources this phenomenon has been analyzed from the perspective of hourly time series. As shown in Figure 14 (for wind) and Figure 15 (for solar), with growing distance, the coefficient of correlation starts to decrease. The drop in correlation coefficient is significantly greater for wind resources than for solar resources. However, at the same time, for wind resources the scatter plot is more dispersed than the one for solar resources. This is because the wind is often a very local phenomenon, especially in a case

of a complex orography, whereas solar resources have a more global nature (apart from cloud movement).

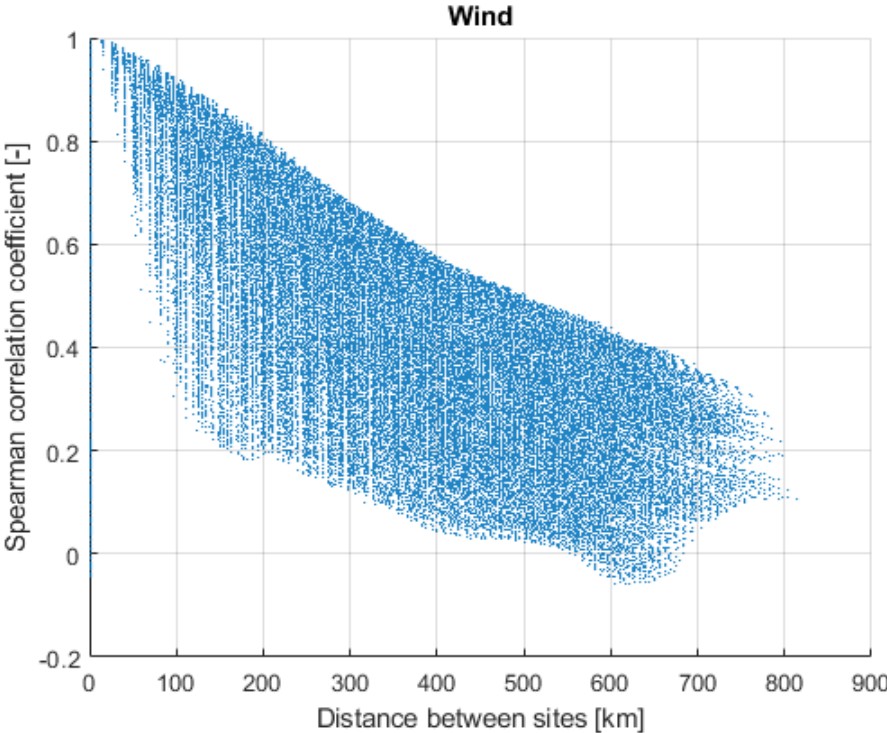

**Figure 14.** Spatial smoothing (decorrelation) for wind resources across Poland based on hourly time series, 1980–2019.

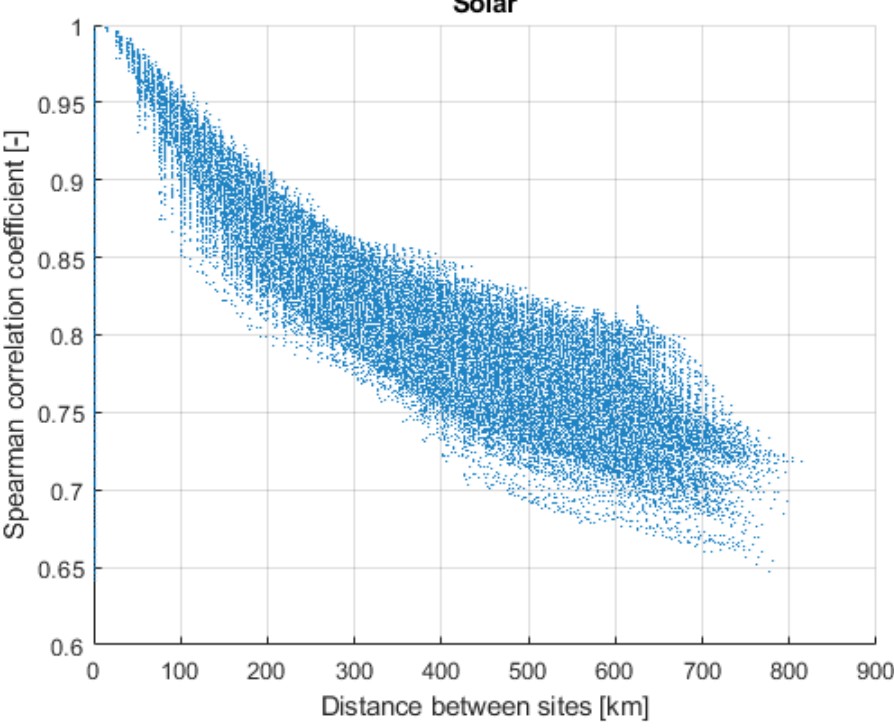

**Figure 15.** Spatial smoothing (decorrelation) for solar resources across Poland based on hourly time series, 1980–2019.

The spatial smoothing effect shown here supports the idea of developing distributed solar and wind generation, but this has to be followed up by the development of transmission infrastructure. A perfect spatial complementarity situation occurs when the correlation between given resources observed on an hourly time scale would be −1. In such case power plants with combined capacity of X MW placed in two locations A and B would be generating power without hourly variability. If serving loads in a power system without transmission capacity constraints, such power plants could provide baseload power without any support. In a case where the correlation never reaches zero, its values observed are lower than 1 one can expect a smoother generation profile, with smaller ramp rates. Then again, this can be exploited only if sufficient transmission capacity is available. Naturally, a trade-off has to be made between selecting sites with the most promising wind resources, on the one hand, and distributing wind parks to benefit from spatial smoothing, on the other. More smooth generation patterns can be obtained by distributing renewable power plants across greater area but simultaneously this could be achieved at a cost of sitting power plants in locations characterized by poorer resources.

*3.4. 'Resource Droughts': Frequency and Duration*

The earlier sections discussed the solar and energy resources' potential, spatial and temporal complementarity and spatial smoothing effect. This section investigates the frequency of 'resource droughts' as defined in Section 2. This phenomenon was analyzed from the perspective of daily time series. For each point, the occurrence of 'resource droughts' has been identified using averaged national daily generation as a reference.

Figure 16 shows the spatial distribution of the probability of 'resource droughts' locally (left column) and nationally (right column). The solar resources show a clear division in terms of 'resource droughts' between a region of higher probability (northern Poland) and one of lower probability (southern Poland). This is broadly in line with resource availability (Figure 3), but it is interesting that, even from the local perspective, a region with fewer solar resources has a higher number of droughts. A slightly contrary phenomenon is observed for wind resources. Here, from the local perspective, a region (Figure 2) with high resource availability (northern Poland) and a region with significantly lower availability (southern Poland) both have a high probability of 'resource drought' occurrence. When both resources are analyzed in combination (Figure 16, lower two maps), the highest probability of 'resource droughts' occurs in the northern half of Poland (excluding the western and central stretch of the Baltic coast).

Figures 17 and 18 show the results of 'resource drought' analysis for both locations. The analysis focuses on the regional (left column) and national perspectives (right column). In the regional analysis, the threshold defining 'resource drought' is calculated based on local conditions, whereas from the national perspective it is calculated as the mean of all locations. As shown in Figure 17, for the Bełchatów location, there is very little difference between the droughts from the local and national perspectives. This indicates that this location is a very good representation of typical Polish wind and solar resources. As expected, the majority of 'solar droughts' occur in the autumn–winter period, whereas the opposite is true for wind resources. However, 'wind droughts' seem to be more evenly distributed over the year. When both resources are considered jointly, the number of droughts decreases, but they are still persistent during the autumn–winter period. Apart from this, we also determined that in the majority of locations in Poland (over 80%) there are no statistically significant trends in 'resource drought' intensity.

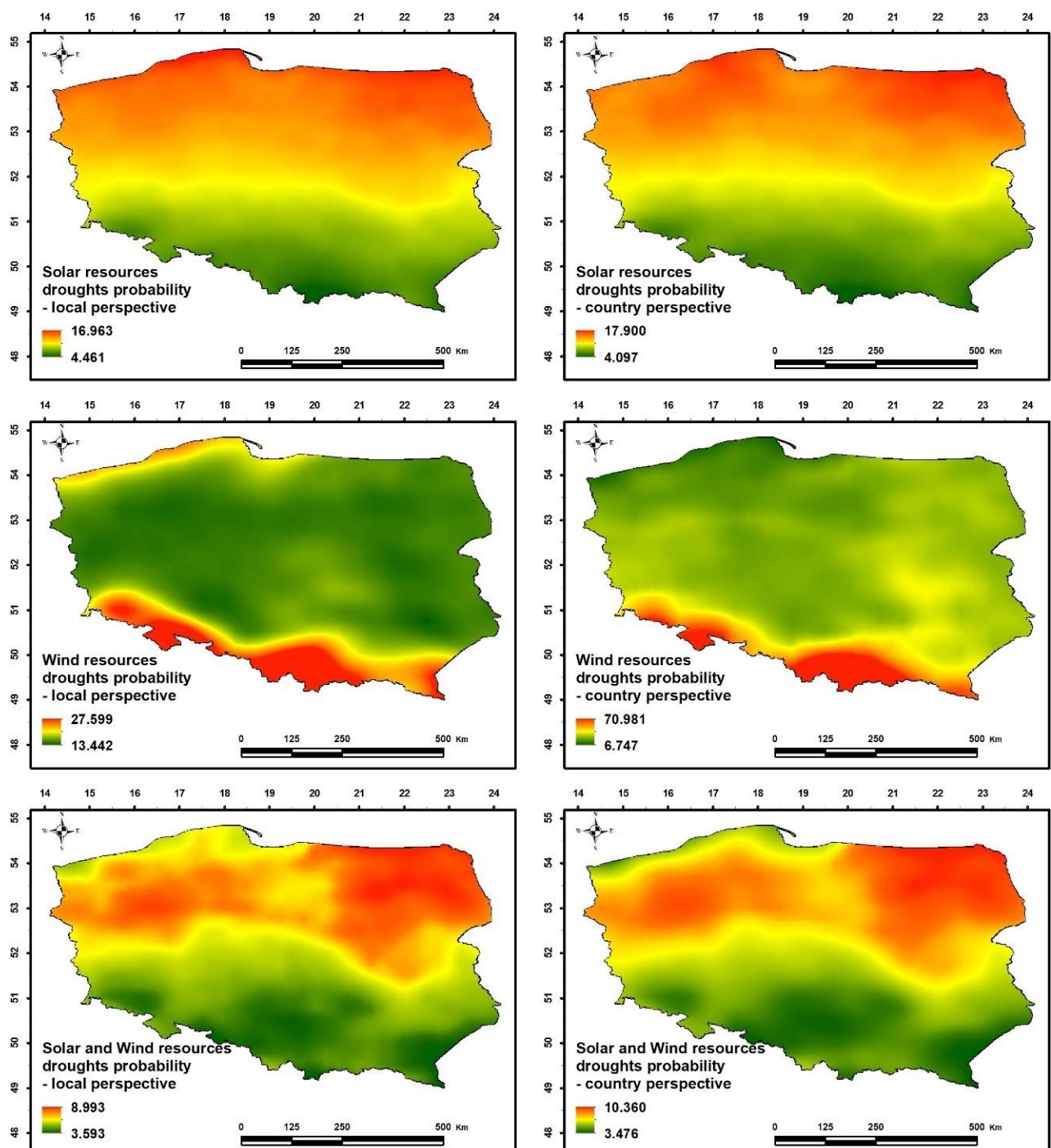

**Figure 16.** Probability [%] of local and national 'resources droughts' based on daily sums of resources availability, 1980–2019
To better represent 'resources drought' events, two locations have been considered, one in central Poland near the Bełchatów (51.2° N, 19.4° E) brown coal power station and open-pit mine, and a second in Łeba on the Baltic coast (54.6° N, 17.50° E). The selection of locations was not random. First, we selected the Bełchatów region because in 2030 the owner will probably begin phasing out the power station blocks. This location is very favorable for developing other power stations due to its location in central Poland and its existing strong transmission capacity (Bełchatów brown coal power station's nameplate capacity exceeds 5 GW). The second location is characterized by one of the best wind resources; Pomerania voivodeship (where Łeba is located) also has one of Poland's highest installed capacities in wind generation.

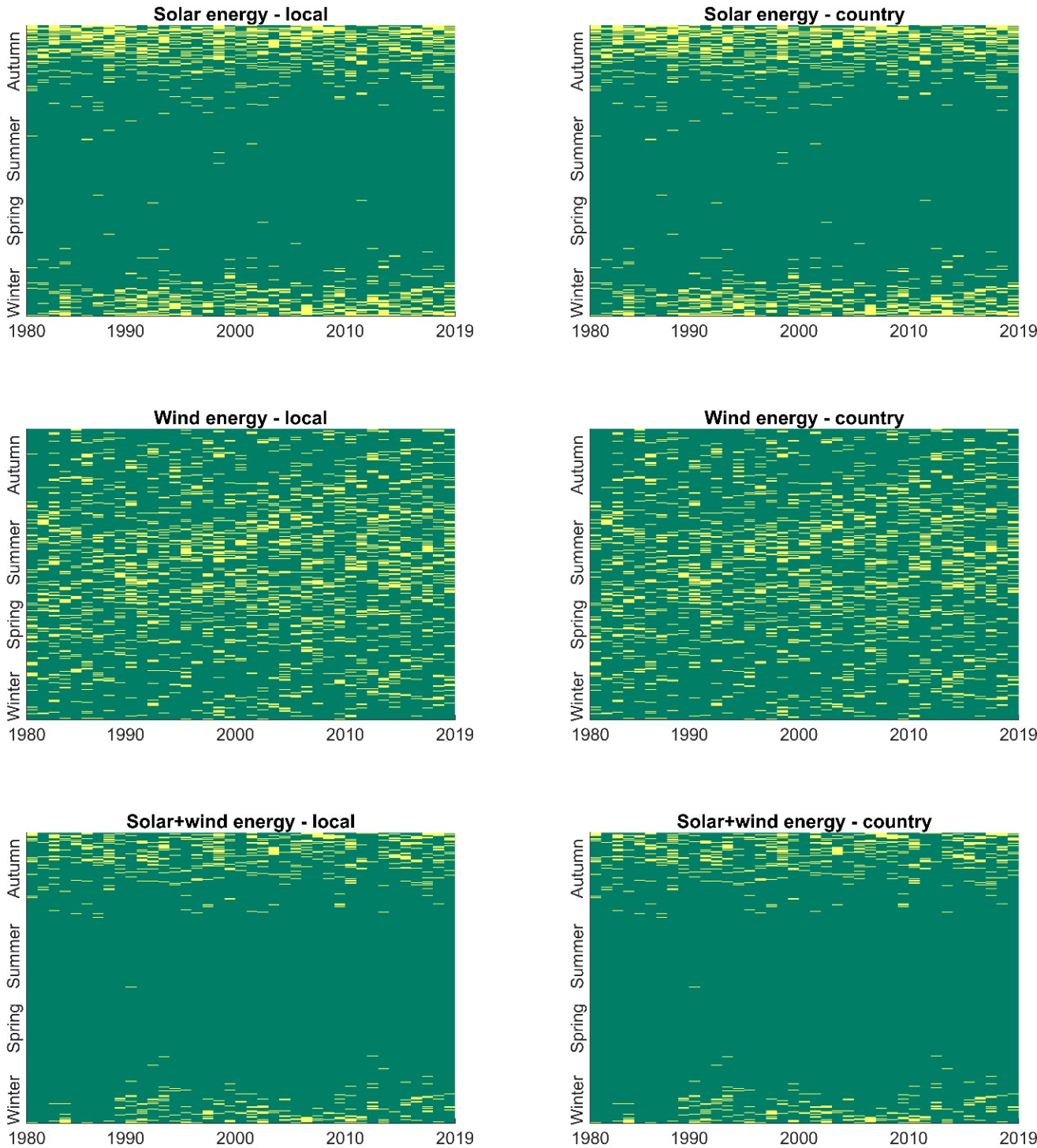

**Figure 17.** 'Resource droughts' (yellow) in Bełchatów region (central Poland). Left column presents local 'resource droughts', where the threshold for classifying a day as a 'resource drought' event was based on mean daily resources availability over the years 1980–2019. Right column presents a similar analysis, but with the threshold based on national mean (taking into account all grid points within Polish borders).

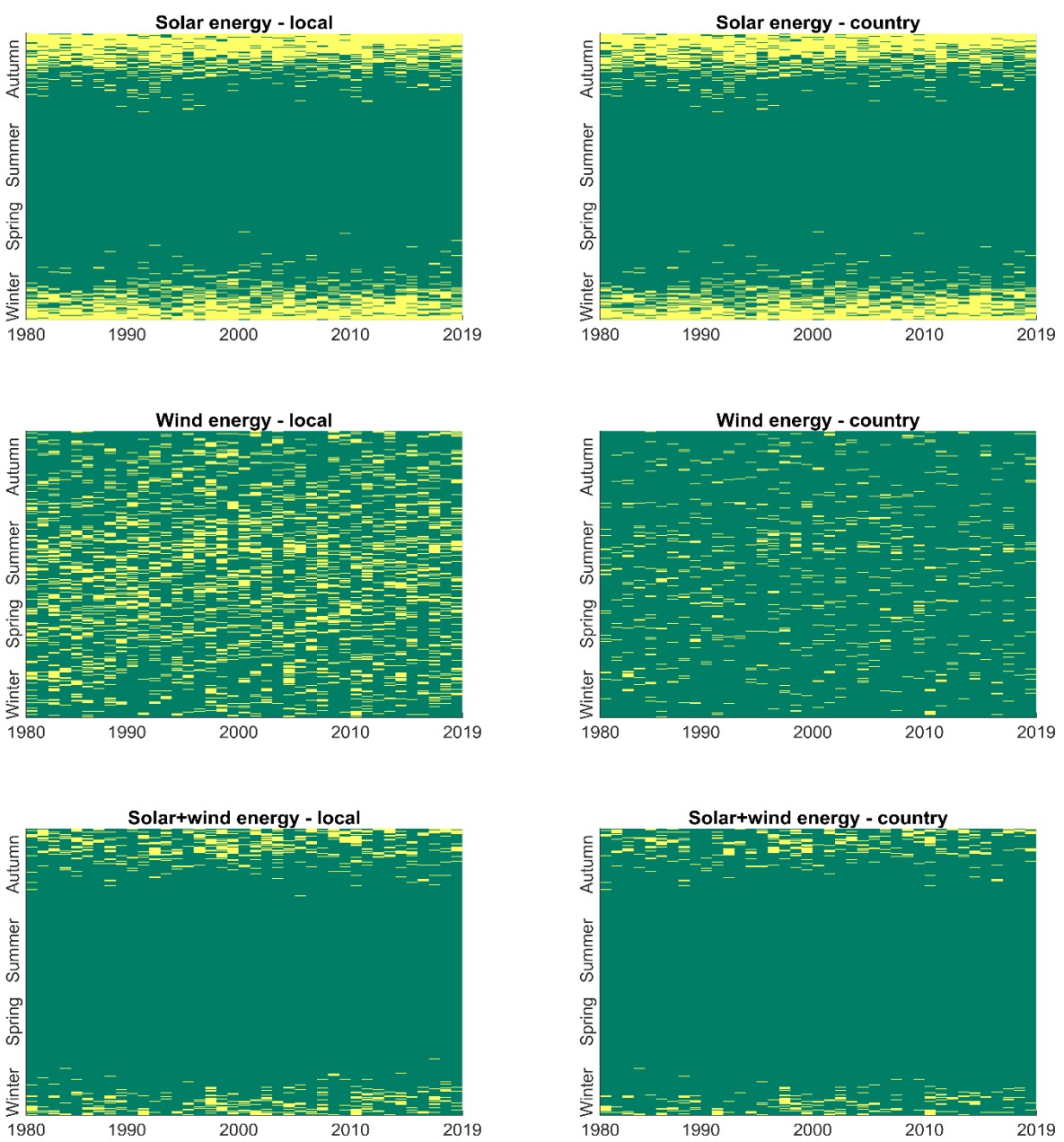

**Figure 18.** 'Resource droughts' (yellow) in Łeba region (central Poland). The left column presents local 'resource droughts', where the threshold for classifying a day as a 'resource drought' event was based on mean daily resources availability over the years 1980–2019. The right column presents a similar analysis, but with the threshold based on national mean (taking into account all grid points within Polish borders).

A similar analysis has been performed for the Łeba location. It was found that in terms of solar resources it exhibits exactly the same pattern, whether from the local or country-wide perspective. However, when it comes to wind resources, the number of national droughts is much smaller. This is quite obvious, since this site has greater wind resources than the rest of Poland. However, even though the wind resources are significantly more abundant in Łeba than in Bełchatów, the number of local 'resource droughts' is significantly higher. For Łeba, over the period of 40 years, 3049 days were identified that could be considered as 'resource drought', whereas for Bełchatów the figure was 2218, which correspond to 11.6%, and 8.4%, respectively. This indicates that, although wind resources in Łeba are much more abundant than the national average, they are also more variable. Nevertheless, if we compare both locations to the country-wide figures for

joint solar–wind resources, we see that the occurrence of 'resource droughts' is slightly smaller in Łeba (570 days over the 1980–2019 period) compared to Bełchatów (706 days over the 1980–2019 period). This indicates that, from the power system point of view, a hybrid power station could ensure a more reliable power supply if located in Łeba. Still, in such an analysis, one also has to consider other factors, like the relation to the load [31] or availability of transmission capacity and resulting curtailment of wind power [58,59].

## 4. Conclusions

In this paper, the complementarity of solar and wind resources has been assessed based on ERA5 data, covering the years 1980–2019. The Spearman coefficient of correlation was used to assess complementarity. For the first time in the literature the complementarity for those two energy sources has been presented for Poland in the form of complementarity maps. The phenomenon of 'resource droughts' has been analyzed from both local and national perspectives. It was found that regions characterized by stronger wind energy potential may suffer from more frequent local 'resource droughts'. In terms of solar resources, we observed an increasing trend in terms of availability; for wind, the opposite was found, but statistically significant trends were identified only for central Poland. This finding is particularly important in the light of the observed increasing temperatures and their impact, both on the performance of renewables and on energy demand [60]. Complementarity analysis showed a good complementarity between resources on the monthly timescale, whereas for daily and hourly scales some negative correlation exists, but at significantly lower levels (less than $-0.35$). On the country level, the solar and wind resources tend to complement each other on an inter-annual perspective, although this might not be true for all locations.

In the light of the conducted analysis, we summarize that this work provides meaningful tools and methods for more detailed analysis of solar and wind energy resources variability and complementarity, and of their potential role in the power system.

Many potential future research directions have been identified. Firstly, there is a need to investigate how the availability of local resources matches energy demand (on various time scales) and to analyze the future potential need to expand the transmission infrastructure [61]. Secondly, we would like to emphasize the need to create maps with guidelines indicating the optimal configuration of off-grid energy systems that consider the complementarity and variability of local energy sources. Finally, solar and wind are not the only variable renewable energy sources in Poland, and future works should also consider the complementarity of the solar–wind–hydropower trio (which is also important even for lowland countries), since they too have the potential to develop small-scale hydropower projects [62,63]. This is especially important as there is a pressing need to efficiently utilize available water resources to overcome droughts [64,65] and to improve the water–energy–food nexus [66,67].

**Author Contributions:** J.J., Conceptualization, methodology, statistical analysis, software, validation, formal analysis, investigation, resources, writing—original draft preparation, writing—review and editing, visualization, supervision, project administration. J.M., conceptualization, investigation, writing—review and editing, funding acquisition. M.G., conceptualization, methodology, software, validation, formal analysis, writing—original draft preparation, writing—review and editing, visualization. P.B.D., software, formal analysis, data curation, writing—review and editing, B.K., methodology, statistical analysis, software, validation, data curation, writing—review and editing. All authors have read and agreed to the published version of the manuscript.

**Funding:** Publication financed by subsidies for the maintenance and development of research potential of AGH in Cracow. Wydanie publikacji finansowane z subwencji na utrzymanie i rozwój potencjału badawczego AGH w Krakowie.

**Institutional Review Board Statement:** Not applicable.

**Informed Consent Statement:** Not applicable.

**Data Availability Statement:** Data used in this study is publicly available at ref. [32].

**Conflicts of Interest:** The authors declare no conflict of interest.

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
