# Peer review of "Complementarity and ‘Resource Droughts’ of Solar and Wind Energy in Poland: An ERA5-Based Analysis"

_energies, doi:10.3390/en14041118_

Round 1
Reviewer 1 Report
This paper analyzed the spatial and temporal behavior of solar and wind resources based on reanalysis datasets from ERA5. The research topic is interesting and the structure of the paper is relatively complete. However, I have the following major concerns.
- The language should be double checked, there are some grammar mistakes, such as, “a n adequate spatial” in the abstract.
- The abstract should be well organized, the main novelty, methodology, objective and significance of the study should be clearly clarified. The novelty and methodology are not quite clearly described at the present.
- The assumptions, boundary conditions and solution method should clearly clarified.
- The author contributions should be clearly clarified.
Author Response
Dear Reviewer,
Thank you for your suggestions. Please see attached our responses to your comments and also to the remaining reviewers.

Reviewer 2 Report
The author’s present an analysis of solar and wind resources over Poland, with special interest on the complementarity between these resources. It is an interesting topic that has been investigated from different perspectives, but it is interesting to assess complementarity in different areas.
However, I have detected issues in the presented manuscript that must be addressed:
First, I don’t think that a complementarity analysis could be made using just one year of data. Although solar and wind resources have more variability in daily and seasonal scales, it is important to remark that inter-annual variability also exists and in order to present robust results, it could be necessary to consider longer time scales. One of the main advantages that the using of reanalysis data has, is that the time series is long enough to do climatological studies. For this reason, I think that the authors should consider make an analysis including more years to have robust results. In addition, it is known that the 2018 was specially sunny in many regions of Central Europe: (see this report from Copernicus https://climate.copernicus.eu/sunshine-duration), so the results obtained might not be representative of the mean climate.
I also have found that in some cases it is used ‘energy droughts’ instead of ‘resource droughts’ in the text, as if they were equivalent. This could be misleading for the reader. Regarding that, part of the analysis is made showing relationships between resources and electricity demand, but it does not seem to be the objective of the paper, as it is not mentioned either in the abstract or introduction.
About the relationship between resources and electricity demand, I would add that both resources have a seasonal pattern, which is something expected. If the electricity demand has not a seasonal pattern, it is normal that the correlation between resources and demand is very low. In the daily time scale, it is not shown how it is the electricity demand curve, but if it is similar to other European countries, probably it would have a (second) maximum in the evening, that cannot be supplied with solar energy, so it is something already expected. However, it would be more interesting to analyse the daily wind curve, not only aggregated, but at different zones. Also, at figure 3, ‘work’ days or holidays are mixed, so is it possible that the relationship between wind and demand could be biased?.
More specific comments:
Abstract:
- Line18: correct ‘a n’
- Line 24-26: energy droughts is not defined and should be called ‘resource droughts’
Introduction:
- Line 56-57: Please rewrite the sentence: ‘In many countries…”
Materials and methods:
- Line 69: include the name of the dataset or reanalysis, not only the reference.
- Line 70-71: if the resolution is 0.1º, the reanalysis is ERA5-land, which is not the same as ERA5.
- Line 83: move the reference [27] where the reading is not interrupted.
- Line 100: it is not clear if correlation is local or for the whole country.
- Line 110: include ‘has’ before been.
- Line 108-136: I found this explanation unnecesary.
- Line 139: There is an extra withe space between ‘to’ and ‘analyze’
Results:
-Line189: If I am not wrong, it is not explained in the methodology from which dataset is obtained the demand data.
- Line 191: The representation from figure 3 is not supply, they are resources.
- Line 200: In the sentence “The additional…” include the time scale.
- Line 227: correct ‘expresses’
- Line 230: you say “significant spatial variation in Poland”, but it is very difficult to see that with the color scale of the figure.
- Figure 4: Maybe is better to choose the same scale for all the figures, to make easier the comparison.
- Line 268: Could you please give the standard deviation and mean data to obtain the CV?.
- Line 272: It is not still a large CV even with combined resources?
- Figure 5: Is it hourly or daily scale? If it is hourly, include daily.
- Line 291: You should say ‘resource droughts’.
- Figure 6: You should include droughts from wind and solar separately, in order to compare with the combination.
- Line 309: correct ‘mich’.
- Figure 7: Explain the figure in the caption.
Conclusions:
- Line 320 (and everywhere else): change energy droughts, they are resource droughts.
- Line 326: longer time series are available for ERA5 reanalysis, so the work should be done using them. Results for 2018 might not be robust enough. In addition, it is interesting to see also inter-annual variability.
Author Response
Dear Reviewer,
Thank you for your suggestions. Please see attached our responses to your comments. We are also attaching responses to the remaining reviewers.

Reviewer 3 Report
This manuscript analyzes the complementarity between solar and wind energy, and applies this to the Polish context. On top of this, the paper discusses the impact of resource droughts. Although the study seems to be executed well, I am left a bit puzzled when finished reading it. What did I learn from this manuscript? This remains very unclear. Some concerns and asuggestions:
- The introduction should better position the manuscript within the literature. What is the contribution? What scholarly debate the authors want to engage in? This is especially true as recently several review articles on complementarity were published.
- Why do we need to study Poland? What makes this country so interesting to study this topic in this context?
- Moving from section 1 to section 2 feels like missing out on an important part of the text. The authors discuss input data, while the reader still has no idea what the goal, design and setting of the empirical study involves.
- I would like to see a strong discussion why we need to just look at 2018 and not at other years. How can we measure seasonal effect if we just look at one year? Are the seasons in 2018 typical, below or above average to a long-term benchmark?
- Why do they use ERA5 data? What are alternative sources?
- I missed a table with descriptive data in section 2.1.
- The difference between potential and generation (lines 90-91) remains unclear as well as the impact of this choice on the study results.
- The empirical analysis is just based on a correlation coefficient. Do the review articles [13-16] not offer more sophisticated models?
- Why discuss droughts in this paper?
- Section 3.2: unclear how many data points are included.
Author Response

(The authors gave the same response as above.)

Round 2
Reviewer 2 Report
I want to thank the authors because of the effort to accomplish the reviewers requirements for improving the manuscript.
Before accepting the paper for publication it is necessary to address the next issues:
- It is very interesting to see the long-term trends of the resources. However, I think there is a lack of references to support the results, considering that similar analyses have been made in other works. For instance, I find it necessary to include references on the long-term trends of wind resource over Europe, and the possible explanations. Is this paper showing same results or something different? In terms of solar resource, do you think that the increasing trend could be related with the ‘brightening’ period observed in Europe due to the reduction on atmospheric aerosols?. It is necessary not only to show results, but also to put them in context with the work already done to find discrepancies or agreements.
- On the contrary, I find an exhaustive description of previous works on complementarity. I think that the authors should reduce the introduction and description on these topics and include some comments on solar and wind resource, probably previous work in wider areas, due to the fact that they not only show complementarity results, but also wind and solar themselves.
- The spatial smoothing section is very weak. Only how the correlation decreases is show, but it is purely descriptive and expected, but I think that no conclusion can be made from this section.
Specific comments:
Abstract:
- Line 13: I would change generators by ‘power plants’.
- Line 27: you should briefly describe what you mean with ‘resource drought’ (for example: ‘days below a threshold’).
Introduction:
- Line 103: You need to explain what is ‘energy drought’ as in the abstract, so the reader do not have to go to the reference to look for it.
Materials and Methods:
- I would include an introduction paragraph to explain that you will describe first the data and afterwards each method applied.
- Line 155-175: I still don’t find interesting this explanation about what ‘complementarity’ means.
- Line 186: “spatial smoothing”: it is not clear how yoy apply this to your data. A more detailed explanation on how you obtain aggregated series with gridded data is necessary, and how are afterwards evaluated.
- Line 210: Please, clarify here if trends are calculated for each cell or aggregated series.
Results:
- Line 262-264: is it that trend observed in other studies? What is the reason behind?
- Line 272: You could include the interannual CV.
- Line 337: Fig 7-9, should be 8-10?
- Figures from 7 to 15: Please, correct the figures numbers, there is a mistake because there is 2 “Figure 7”.
- Line 386-388: First, I think it is not very well explained how the Figures 13 and 14 are created (spatial smoothing). I cannot find the explanation either here or in the methodology and it is difficult to understand the graphs. Secondly, you are showing how correlation decreases, but in both cases the minimum is not below zero, should I understand that is that enough? Or is it necessary to have correlation closed to -1 in order to have complementarity?.
- Line 393: ‘amid’
- Line 406: Pleas, correct the text.
- Line 413: Please, explain how are these calculations made.
Results
- Line 484: include quotation or not for ‘energy droughts’, but do the same in each case on the text.
Author Response
Dear Reviewer,
Thank you very much for your comments. Please see attached our responses and actions undertaken with regard to your suggestions.

Reviewer 3 Report
Thank you for sending me the revised version of this article. The authors addresses most of my concerns.
Author Response
Thank you very much for your previous comments.
We have now submitted the revised version of the manuscript accordingly to the new comments from Reviewer #3. Please see our responses to his/her comments.
Round 3
Reviewer 2 Report
Thank you for the improvement of the manuscript. I find it ready to be accepted for publication.